# A Structural Effect of the Antioxidant Curcuminoids on the Aβ(1–42) Amyloid Peptide

**DOI:** 10.3390/antiox14010053

**Published:** 2025-01-05

**Authors:** Angelo Santoro, Antonio Ricci, Manuela Rodriquez, Michela Buonocore, Anna Maria D’Ursi

**Affiliations:** 1Department of Pharmacy, University of Salerno, Via Giovanni Paolo II, 132, 84084 Fisciano, Italy; asantoro@unisa.it; 2Fresenius Kabi iPSUM, Via San Leonardo, 23, 45010 Villadose, Italy; antonio.ricci@fresenius-kabi.com; 3Department of Pharmacy, University of Naples Federico II, Via Domenico Montesano, 49, 80131 Naples, Italy; manuela.rodriquez@unina.it; 4Department of Chemical Sciences and Research Centre on Bioactive Peptides (CIRPEB), University of Naples Federico II, Strada Comunale Cintia, 80126 Naples, Italy

**Keywords:** Alzheimer’s disease, curcumin, antioxidants, endoplasmic reticulum stress, unfolded protein response, circular dichroism, molecular dynamics

## Abstract

Investigating amyloid–β (Aβ) peptides in solution is essential during the initial stages of developing lead compounds that can influence Aβ fibrillation while the peptide is still in a soluble state. The tendency of the Aβ(1–42) peptide to misfold in solution, correlated to the aetiology of Alzheimer’s disease (AD), is one of the main hindrances to characterising its aggregation kinetics in a cell-mimetic environment. Moreover, the Aβ(1–42) aggregation triggers the unfolded protein response (UPR) in the endoplasmic reticulum (ER), leading to cellular dysfunction and multiple cell death modalities, exacerbated by reactive oxygen species (ROS), which damage cellular components and trigger inflammation. Antioxidants like curcumin, a derivative of Curcuma longa, help mitigate ER stress by scavenging ROS and enhancing antioxidant enzymes. Furthermore, evidence in the literature highlights the effect of curcumin on the secondary structure of Aβ(1–42). This explorative study investigates the Aβ(1–42) peptide conformational behaviour in the presence of curcumin and six derivatives using circular dichroism (CD) to explore their interactions with lipid bilayers, potentially preventing aggregate formation. The results suggest that the synthetic tetrahydrocurcumin (THC) derivative interacts with the amyloid peptide in all the systems presented, while cyclocurcumin (CYC) and bisdemethoxycurcumin (BMDC) only interact when the peptide is in a less stable conformation. Molecular dynamics simulations helped visualise the curcuminoids’ effect in an aqueous system and hypothesise the importance of the peptide surface exposition to the solvent, differently modulated by the curcumin derivatives.

## 1. Introduction

Alzheimer’s disease (AD) is a neurodegenerative disease characterised by the progressive accumulation of amyloid–β (Aβ) peptides and hyperphosphorylated tau, which form plaques and neurofibrillary tangles, respectively [1,2]. According to the amyloid cascade hypothesis, two Aβ peptides—Aβ(1–40) and Aβ(1–42)—are produced as soluble molecules but, in response to environmental factors, aggregate into low-molecular-weight soluble oligomers and high-molecular-weight protofibrillar oligomers (PFOs), which, in turn, give rise to insoluble fibrils, eventually forming amyloid plaques [3,4,5].

Endoplasmic reticulum (ER) stress is a pathological condition where the organ is burdened by misfolded or poorly assembled proteins [6,7]. During ER stress, the unfolded protein response (UPR) activates to restore homeostasis by reducing global protein synthesis, increasing the protein folding capacity, and degrading misfolded proteins [8,9].

One of the hallmarks of ER stress is the high production of reactive oxygen species (ROS), damaging proteins, lipids, and DNA, exacerbating cellular stress and triggering inflammatory responses [10,11,12,13,14,15,16].

When ER stress persists and the UPR is chronically activated, it can lead to cellular dysfunction and, in extreme cases, cell death [17,18,19,20,21,22].

Conditions of ER stress have been found in the brains of AD patients, associated with synaptic dysfunction, neuroinflammation, and neuronal death. The environmental conditions of ER stress represent an ideal medium to favour Aβ peptide conformational change and to stimulate the formation of amyloid plaques. Given the high concentration of ROS, antioxidant agents may play a critical role in coping with the dysfunction caused by ER stress. Concerning the Aβ peptide, a great deal of data show that antioxidant molecules may control the transition to β-structures and the formation of amyloid plaques [23,24,25,26].

In several works, we analysed the conformational behaviour of Aβ(1–42) and its shorter fragment Aβ(25–35) in different conditions to identify the environmental physical-chemical properties favouring the transition of the peptides toward β-sheet structures [27,28,29,30,31]. Part of this investigation aimed to explore how antioxidant molecules affect the conformational stability of Aβ peptides and their potential to destabilise lipid cell membranes. We examined Aβ(25–35) in the presence of flavonoids, a group of naturally occurring compounds recognised for their health benefits due to antioxidant properties. Notably, we found that the flavonoids rutin, quercetin, naringin, and naringenin demonstrated a neuroprotective effect by inhibiting the interaction of Aβ(25–35) with cell membranes and its permeation through their lipid bilayer [32].

Research shows that omega-3 fatty acids (FAs) serve as enhancers in the antioxidant defence against ROS. They significantly boost serum total antioxidant capacity (TAC) and glutathione peroxidase (GPx) activity while lowering malondialdehyde levels, thereby enhancing antioxidant status. Our findings reveal that omega-3 FAs play a neuroprotective role by modulating the amyloid peptide’s fate and adjusting the microstructural and dynamic characteristics of the neuronal membrane. Docosahexaenoic acid (DHA) acts as a crucial membrane-fluidising agent, shielding the membrane from damage due to interactions with peptide aggregates and reducing bilayer defects where delipidation begins [33,34].

Among the natural derivatives, curcumin, a polyphenolic compound derived from the root of *Curcuma longa*, exhibits potent antioxidant properties by scavenging ROS [35,36,37,38] while concurrently enhancing the activity of crucial enzymatic antioxidants such as superoxide dismutase (SOD), catalase, glutathione oxidoreductase, and mitochondrial complex enzymes. Additionally, curcumin has demonstrated to have an anti-angiogenic effect by inhibiting the NF-κB pathway [39] and mitigating lipid peroxidation, primarily by diminishing the levels of malondialdehyde (MDA), nitrite, and acetylcholinesterase (AChE) [40,41].

Most notably, curcumin has exhibited an in vitro capability to reduce β-sheet content in amyloid peptides and disaggregate amyloid fibrils [42,43,44], indicating a potential protective effect against AD [45,46,47].

We previously studied the conformational behaviour of Aβ(25–35) in the presence of curcumin and its natural derivatives—demethoxycurcumin (DMC), bisdemethoxycurcumin (BDMC), cyclocurcumin (CYC), and a mixture of them (MIX)—and synthetic derivatives: tetrahydrocurcumin (THC), hexahydrocurcumin (HHC), and octahydrocurcumin (OHC). Supported by molecular modelling, these investigations indicated that curcumin and its derivatives must assume a partially folded conformation to interact with the α-helix form of Aβ(25–35) and prevent the formation of aggregates [48].

In the present study, we report a conformational analysis based on CD spectroscopy of the full-length Aβ(1–42) peptide in the presence of curcumin and the above-mentioned derivatives in lipid and organic systems.

Circular dichroism (CD) serves as a valuable method for quickly assessing the secondary structure and folding of peptides. Its reproducibility enables the observation of conformational changes in peptides under various environmental conditions, as well as their interactions with other molecules [49,50].

Our findings indicate that THC influences the Aβ(1–42) secondary structure across all tested conditions, while OHC, BDMC, and CYC derivatives notably disrupt amyloid conformation under specific environmental factors, such as in the presence of low concentrations of HFIP. We also performed molecular dynamics (MD) simulations to describe the influence of the most interesting curcuminoids resulting from CD analysis (BDMC, CUR, CYC, OHC, and THC) in complex with Aβ(1–42) oligomers in a system composed of pure water at a physiological pH.

This study focuses on how curcumin and its derivatives influence the secondary structure of the peptide while considering the effects of organic or membrane-mimetic systems. The results suggest that these small molecules increase the exposure of Aβ(1–42) oligomers to the solvent, indicating that the conformational changes observed in vitro may be due to a “deshielding” mechanism driven by the curcumin derivatives. This analysis aims to elucidate the peptide’s dynamic behaviour in the presence of potential ligands, with the goal of optimising a conformational-based strategy to modulate protein misfolding and alleviate ER stress and programmed cell death.

## 2. Materials and Methods

### 2.1. Recombinant Aβ(1–42) Peptide Expression

Aβ(1–42) peptide was produced by transforming *E. coli* BL21(DE3)-pLysS cells with the PetSac plasmid, provided by Walsh’s research group [51]. To optimise expression levels, Ca^2+^-competent *E. coli* BL21(DE3)-pLysS cells, prepared by thermal shock, were cultured on agar plates containing Luria Bertani (LB) medium with ampicillin (50 mg/L). Individual colonies, stored at −80 °C, were taken from a stock solution and added to 50 mL of preinoculum containing 50 μL of ampicillin (50 mg/L). This preinoculum was incubated overnight at 37 °C with constant stirring. The next day, 5 mL of this culture was transferred to 500 mL of LB medium with 500 μL of ampicillin (50 mg/L). When the cell density reached an OD_600_ of 0.6 at 37 °C, gene expression was induced. After 4 h of induction, the cells were collected and centrifuged to separate the supernatant from the pellet. The pellet was thawed and sonicated three times in a solution containing 10 mM Tris/HCl (pH 8.0) and 1 mM EDTA, for 5 min on ice (using a 1/2 horn at a 50% duty cycle). The sonicated mixture was centrifuged for 10 min at 8000 rcf, the supernatant was discarded, and the pellet was resuspended in a solution of 8 M urea, 10 mM Tris/HCl (pH 8.0), and 1 mM EDTA. This suspension was sonicated as before. The resulting solution, containing Aβ(1–42) in urea-soluble inclusion bodies, was diluted with 10 mM Tris/HCl (pH 8.0) and 1 mM EDTA, and purified using a HiTrap^®^ DEAE column at a flow rate of 1 mL/min with an AKTA purification system. The protein was eluted with a buffer containing 8 M urea, 10 mM Tris/HCl (pH 8.0), 1 mM EDTA, and 1 M NaCl. The eluted fraction was dialyzed against 10 mM Tris/HCl (pH 8.0) and freeze-dried. The peptide’s purity was confirmed using SDS-PAGE with Coomassie Blue staining. SDS-PAGE revealed that Aβ(1–42) was present in a band between 4 and 5 kDa.

### 2.2. CD Sample Preparation

Aβ(1–42) has a strong tendency to form fibrils. Aβ(1–42) was subjected to the procedure described by Jao et al. [52] to keep the peptide in its monomeric form and prevent the formation of oligomers and polymers during experiments. The peptide was initially dissolved in trifluoroacetic acid (TFA) until fully solubilised and left in TFA for three hours. Afterward, TFA was removed using a nitrogen flow, and water was added to the sample for freeze-drying. This defibrillating treatment was conducted just prior to dissolving the peptide in the appropriate solvent (both HFIP/water solution and small unilamellar vesicles, SUVs). The SUVs were composed of pure DOPC or DOPG lipids and were prepared by dissolving 200 μM of each phospholipid in a dichloromethane/methanol (CH_2_Cl_2_/MeOH) solution. This solution was then transferred to a round-bottomed tube, where a thin lipid film was produced by evaporating the solvent using a stream of dry nitrogen [48]. Subsequently, the samples were hydrated by adding an appropriate 10 mM phosphate buffer volume at pH 7.4. The resulting mixture was repeatedly vortexed until a homogeneous suspension was obtained. This method generates a suspension of multilamellar vesicles (MLVs), which is then sonicated for 5 to 10 min to produce small unilamellar vesicles (SUVs).

### 2.3. CD Experiments

The CD spectra were recorded using a JASCO J-810 spectropolarimeter, Oklahoma City, OK, USA, utilising a 1 cm quartz cell at a temperature of 25 °C. Each spectrum was obtained by averaging 4 scans within the wavelength range of 260–190 nm, with a bandwidth of 1 nm and a scanning speed of 10 nm/min. The solvent spectrum was subtracted from each measurement to process the spectra. To record the CD experiments, Aβ(1–42) peptide, previously treated according to the procedure described above, was added to the appropriate solution. All CD spectra were acquired with a final peptide concentration of 0.1 mM and a peptide–ligand molar ratio of 1/5. All spectra were acquired in triplicate, and the mean and standard deviation were calculated. The CD curve deconvolution was performed using the CONTIN algorithms available on the DICHROWEB online platform [53]. Statistical analysis for each system was conducted using the one-way Analysis of Variance (ANOVA) method and the post hoc Dunnett’s test by comparing the percentage of the secondary structure obtained from the deconvolution of triplicates against the control (Aβ(1–42) without any ligand) to determine the statistical significance of the differences [54].

### 2.4. MD Simulations

Five extended structures of Aβ(1–42) peptide were sampled and simulated for three 100 ns long simulations in explicit TIP4P water using the force field CHARMM36 [55] at physiological pH (7.4) and 300 K. The system was neutralised by adding neutralising Na^+^ and Cl^−^ ions. Simulations with the curcumin derivatives were performed by sampling the Aβ(1–42) peptides (i) in extended conformations and randomly distributed in the box, and (ii) as an aggregate resulting from the last step of one of the ligand-free simulations; the 3D structures and the topologies of curcuminoids were prepared using the CGenff web server [56,57]. The energy minimisation of the system was performed, and then the system was equilibrated with NVT and NpT runs. The system temperature and pressure were kept constant at 300 K and 1.01325 bar using the Berendsen weak coupling method [58,59]. The results were used for MD simulations using Particle Mesh Ewald for long-range electrostatics under NpT conditions [60]. The trajectory files were fitted in the box and converted into PDB coordinates using the *trjconv* tool from the GROMACS Package. Snapshots of the run were saved every 10 ns. The secondary structure of the peptides was predicted using the *dssp* tool in GROMACS. A solvent-accessible surface area plot was calculated using the *sasa* tool in GROMACS. The structures were visualised using Maestro 12.9.123 by Schrödinger [61].

## 3. Results

### 3.1. Circular Dichroism

The deconvolution of CD spectra acquired in DOPC indicates that Aβ(1–42) exhibits a significant amount of β-sheet secondary structure, a trait noted in the literature as being more susceptible to aggregation (Figure 1) [3]. However, in the presence of OHC and THC, the β-sheet contribution decreases from 50.4% to 38.3% and 44.9%, respectively, in favour of an increasing percentage of α-helix from 15.2% to 32.4% and 25.5%, respectively, suggesting the higher stability of the amyloid peptide in the presence of these two curcumin derivatives.

Similarly, the CD analysis suggests that in the DOPG system Aβ(1–42) mainly adopts a β-sheet conformation (Figure 2). Also in this case, the presence of OHC and THC derivatives tends to be associated with a decrease in the β-sheet secondary structure from 50.5% to 39.1% and 39.6%, respectively. Furthermore, the presence of CYC is associated with a slight decrease in the β-sheet contribution from 50.5% to 41.0%.

The use of organofluorinated solvents has consistently raised critical issues related to the solvent’s tendency to increase the helicity of peptide samples. The CD spectra, in fact, indicate that Aβ(1–42) adopts a complete helix conformation in the HFIP/water 80/20 *v*/*v* solvent system; however, the presence of curcumin derivatives appears to slightly decrease the helical content in favour of the β-sheet contribution. Remarkably, the presence of THC is linked to a significant disturbance in the secondary structure of the amyloid peptide, with a reduction in helix content from 99.9% to 43.2% and an increase in β-sheet content from 0.0% to 34.2% (Figure 3).

By reducing the HFIP content in the system composed of HFIP/H_2_O 50/50 *v*/*v*, we observe a modest loss of helical percentage across all samples (Figure 4). However, once again, the effect intensifies with THC, causing the helix contribution decrease to 43.0%, while the β-sheet and random coil content increase to 28.0% and 29.0%, respectively.

Finally, we studied the behaviour of Aβ(1–42) by further increasing the percentage of water (HFIP/H_2_O 20/80 *v*/*v*). It was previously seen that this system contains the minimum amount of HFIP to prevent the Aβ transition from helix to β-sheet [62]. The CD spectra acquired in this system show that not only THC, but also BDMC and CYC appear to be involved in an interaction with Aβ(1–42), identifiable by the decrease in the helix contribution from 89.6% to 29.9%, 33.5%, and 40.2%, respectively, and an increase in the percentage of the β-sheet secondary structure, which shifts from 10.4% to 38.5%, 39.4%, and 32.3%, respectively (Figure 5).

### 3.2. Molecular Dynamics

Classical all-atom MD simulations were conducted to examine the effect of curcuminoids on the secondary structure of Aβ(1–42) in an unstable system consisting of pure water at a physiological pH (7.4). Accordingly, three independent 100 ns simulations were carried out on five monomers of the Aβ(1–42) peptide, sampled in an extended conformation, to study the behaviour of the peptide in water as a reference. As expected, all three simulations display the rapid formation of small aggregates comprising the five monomers in largely disordered conformations (Figure 6A–C); however, the analysis of the secondary structure assumed by the single monomers reports an high frequency of helix formation in the central moiety (^12^V-G^25^). This helix has been experimentally detected in the 3D structure of Aβ(1–42) obtained by nuclear magnetic resonance (NMR) in the metastable system composed of HFIP/H_2_O 50/50 *v*/*v* [27], pointing out the importance of this moiety in the early stages of the transition from unstable soluble oligomers to insoluble aggregates (Figure 6D–E). Similarly remarkable is the presence of a β-sheet in the C-terminus (^35^M-I^41^, Figure 6A), whose formation is a crucial step for the amyloid seeding that eventually leads to the formation of the fibrils [63]. Therefore, the comparison of the simulation with the experimental results suggests that the aggregates in the MD are soluble but relatively unstable and close to the early stages of fibrillar formation.

We previously reported that BDMC, CUR, CYC, OHC, and THC are the most effective in affecting the conformation of Aβ(1–42) according to CD spectroscopy analysis. Based on the results of the experimental analyses, we performed 100 ns MD simulations on five Aβ(1–42) monomers in the presence of the above-mentioned curcuminoid derivatives in a 1/5 ratio (five peptides and twenty-five curcumin derivatives). For each curcumin derivative, two simulations were conducted. In the first, the five Aβ(1–42) monomers were randomly dispersed in solution (“unbiased simulation”); in the second, the Aβ(1–42) peptides were sampled as disordered aggregates as resulting from the earlier simulations reported in Figure 6A (“biased simulation”). This second system was investigated to determine whether the small molecules could promote disaggregation. The results reveal the impact of curcumin derivatives on the secondary structure of amyloid monomers and their exposure to the solvent. Figure 7A shows that all the curcumin derivatives have a modest effect in reducing the contribution of the β structures in the peptides and slightly enhancing the helix structures in the “unbiased simulation”; however, when the aggregate is already formed (“biased simulation”), the curcuminoids show no effect on the secondary structure, nor on the disaggregation. Regarding the solvent-accessible surface area analysis, we only present the findings from the “unbiased simulation”, as the “biased simulation” showed no noticeable effects. Accordingly, the curcumin derivatives generally increase the total area of Aβ(1–42) peptides—i.e., the five monomers considered as a whole molecule—that is exposed to the solvent, compared to the simulations that do not include them (Figure 7C). By closely examining the solvent exposure of each Aβ(1–42) monomer, we can see that CUR, OHC, and THC significantly enhance it. Conversely, CYC, the only non-planar derivative, exhibits the least impact, indicating that planarity may be crucial for this purpose (Figure 7D).

## 4. Discussion

Monitoring the behaviour of amyloid peptides in solution is crucial to prevent their aggregation and the formation of neurotoxic plaques. In this work, we used CD spectroscopy to explore the secondary structures assumed by Aβ(1–42) in several membrane mimetic systems in the presence of potential natural ligands derived from *Curcuma longa*. Preliminary studies were conducted in DOPC and DOPG liposomal systems on Aβ(1–42) in the absence and presence of curcumin and its derivatives. The analysis of the CD spectra showed that, in the DOPC system, the only curcumin derivatives showing an effect on the secondary structure of the amyloid peptide are the two synthetic derivatives OHC and THC (Figure 1); in the presence of DOPG liposomes, in addition to OHC and THC, CYC also induces an increase in helix conformation (Figure 2). Using the initial structural information of Aβ(1–42) in HFIP/water mixtures [62,64], we observed the conformational behaviour of the amyloid peptide in these systems. It is interesting to note that in the soluble, highly stable HFIP/H_2_O 80/20 *v*/*v* system, Aβ(1–42) assumes a full helical conformation that is generally perturbed by the presence of the curcuminoids; in particular, the synthetic curcuminoid THC seems to induce a significant reduction in helix content and an increase in β-sheet/random coil secondary structures, potentially deriving from Aβ(1–42)-THC interaction (Figure 4).

By increasing the water percentage—HFIP/H_2_O 20/80 *v*/*v* mixture—the interaction with THC is preserved, and CYC and BDMC also perturb the helix conformation of the amyloid peptide. These preliminary data indicate that the most evident conformational changes in response to curcumin derivative addition are observed when the peptide is in a less stable condition. Notably, except for the synthetic THC derivative, which seems capable of interacting with the amyloid peptide across all systems studied, the impact of CYC and BMDC, and, to a lesser degree, CUR and OHC, on Aβ(1–42) conformation seems to depend on the surrounding environment. Additionally, we suggest a simulated model illustrating the interaction of curcumin derivatives with five monomers of Aβ(1–42) at a 5/1 ratio (twenty-five small molecules to five peptides). The curcumin derivatives (specifically BDMC, CUR, CYC, THC, and OHC) that demonstrated a greater impact on perturbing the secondary structure of Aβ(1–42) in CD studies were chosen for a classical all-atom molecular dynamics (MD) study. The simulations were performed in water at a physiological pH to investigate how these small molecules might impact the amyloid’s conformational shift while excluding external influences. We conducted two simulation sets: one with randomly dispersed initial samples of amyloid peptides (“unbiased simulation”) and the other with partially formed amyloid oligomers (“biased simulation”). The results indicate that curcuminoids do not influence the secondary structure or the solvent-accessible area in the “biased simulation”. This suggests that the complex achieves energetic stability, implying that the ligands do not impact the disaggregation kinetics. These results align with earlier MD simulations demonstrating that Aβ(1–42) aggregation is avoided upon interaction with curcumin and its derivatives, which bind tightly and suppress Aβ nucleation [65]. Conversely, the curcumin derivatives were shown to bind to the already formed Aβ(1–42) fibrils without causing a disaggregating effect, but do prevent further elongation [66,67].

Instead, in the “unbiased” simulation, curcumin derivatives significantly reduce the β-structured content in the peptides while slightly enhancing the contribution of helices. Most importantly, they significantly increase the exposure of Aβ(1–42) surfaces to the solvent (Figure 7)—a finding that has also been previously demonstrated [65]—especially when THC, CUR, and OHC are included. This indicates that the conformational changes detected using CD spectroscopy are primarily influenced by varying exposures to organic and membrane-mimetic systems, with curcumin derivatives potentially enhancing this effect.

In conclusion, the involvement of curcumin in ER stress might depend on its interference with several biochemical pathways, and the data remain insufficient to explain the link between curcumin, ER stress, and neurodegenerative diseases [68]. However, the present structural investigation proposes that the demonstrated neuroprotective effect of curcumin derivatives in AD [45,46,47] might also be related to their direct interaction with Aβ monomers, thus preventing the early stages of peptide oligomerisation. Moreover, we believe that this approach can be adapted for the initial study and assessment of a wide range of misfolded proteins linked to UPR activation during ER stress, which remains challenging to investigate in solution.

## Figures and Tables

**Figure 1 antioxidants-14-00053-f001:**
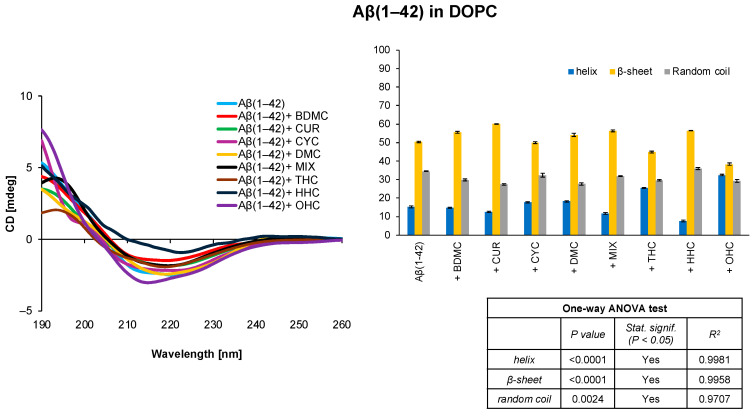
CD spectra and secondary structure quantification performed with CONTIN algorithm of Aβ(1–42) free peptide and in the presence of curcumin derivatives in DOPC SUVs. The results of the statistical analysis conducted using the one-way ANOVA method are reported in the table.

**Figure 2 antioxidants-14-00053-f002:**
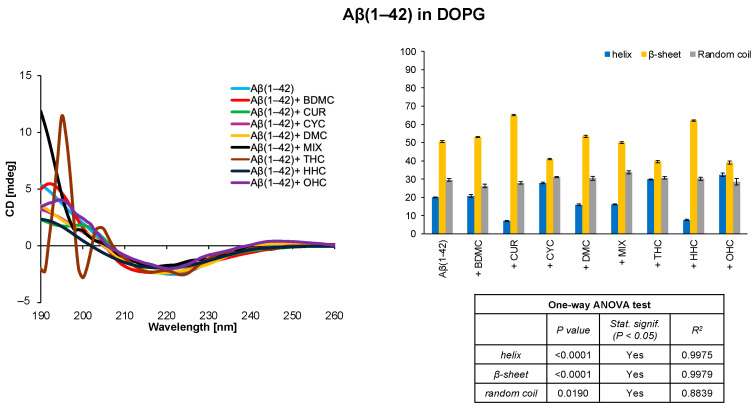
CD spectra and secondary structure quantification performed with CONTIN algorithm of Aβ(1–42) free peptide and in the presence of curcumin derivatives in DOPG liposomal system. The results of the statistical analysis conducted using the one-way ANOVA method are reported in the table.

**Figure 3 antioxidants-14-00053-f003:**
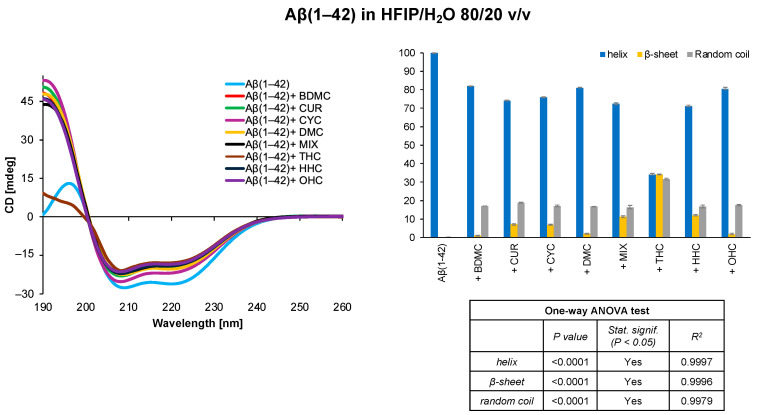
CD spectra and secondary structure quantification performed with CONTIN algorithm of Aβ(1–42) free peptide and in the presence of curcumin derivatives in HFIP/H_2_O 80/20 *v*/*v*. The results of the statistical analysis conducted using the one-way ANOVA method are reported in the table.

**Figure 4 antioxidants-14-00053-f004:**
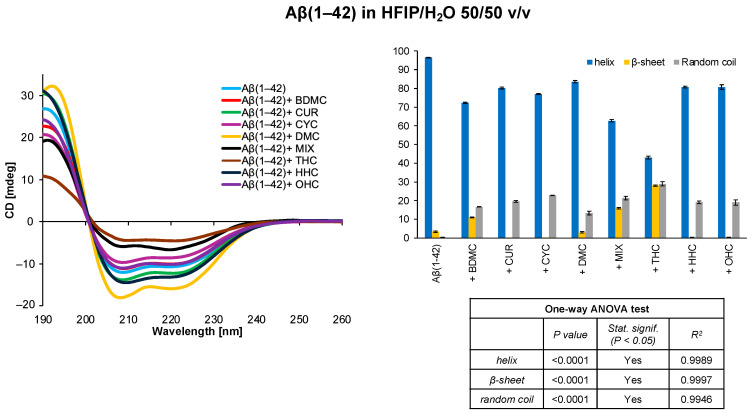
CD spectra and secondary structure quantification performed with CONTIN algorithm of Aβ(1–42) free peptide and in the presence of curcumin derivatives in HFIP/H_2_O 50/50 *v*/*v*. The results of the statistical analysis conducted using the one-way ANOVA method are reported in the table.

**Figure 5 antioxidants-14-00053-f005:**
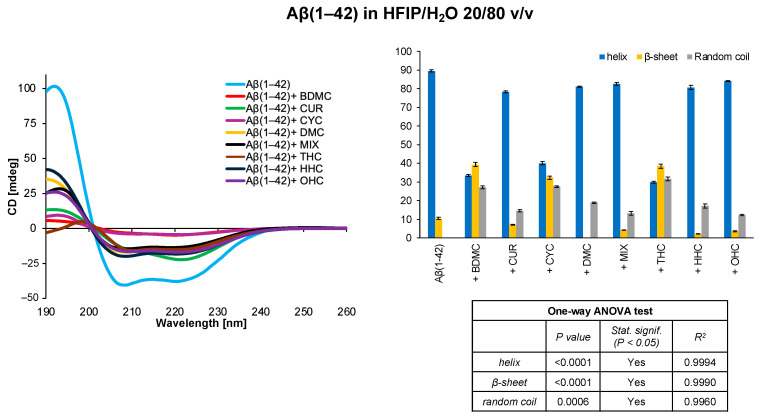
CD spectra and secondary structure quantification performed with CONTIN algorithm of Aβ(1–42) free peptide and in the presence of curcumin derivatives in HFIP/H_2_O 20/80 *v*/*v*. The results of the statistical analysis conducted using the one-way ANOVA method are reported in the table.

**Figure 6 antioxidants-14-00053-f006:**
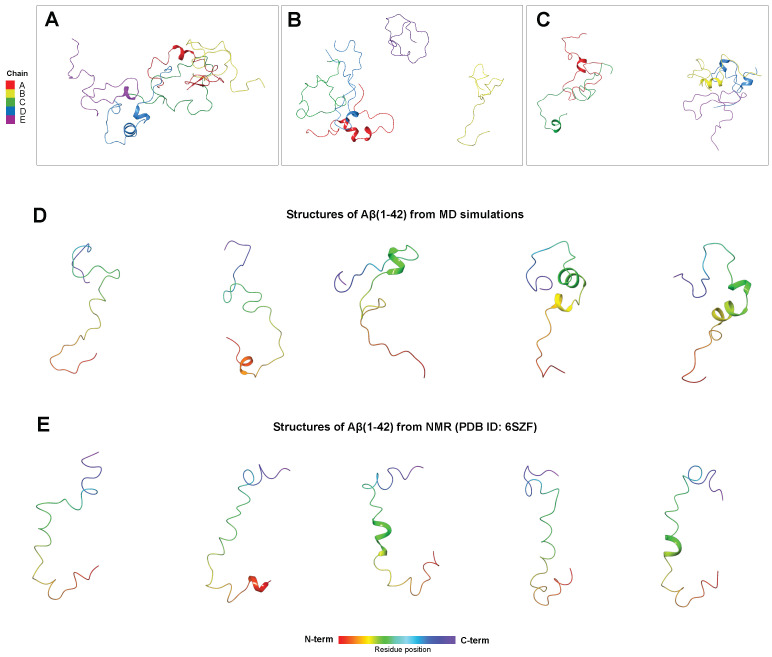
(**A**–**C**) Snapshots of the last steps of the three independent 100 ns MD simulations performed on five Aβ(1–42) monomers in water at pH 7.4. The peptides are reported in ribbon representation and coloured according to the chain name (A–E, red to purple). The α-helix moieties are depicted as wider spiral ribbons, while the β-sheets are indicated as flat arrowed ribbons. (**D**) Selected structures of Aβ(1–42) extracted from the MD simulations compared to (**E**) structures of Aβ(1–42) in the bundle obtained by acquiring 2D NMR spectra in HFIP/H_2_O 50/50 *v*/*v*; the ribbons here are coloured according to the residue position in the peptide (N-terminus to C-terminus, red to purple).

**Figure 7 antioxidants-14-00053-f007:**
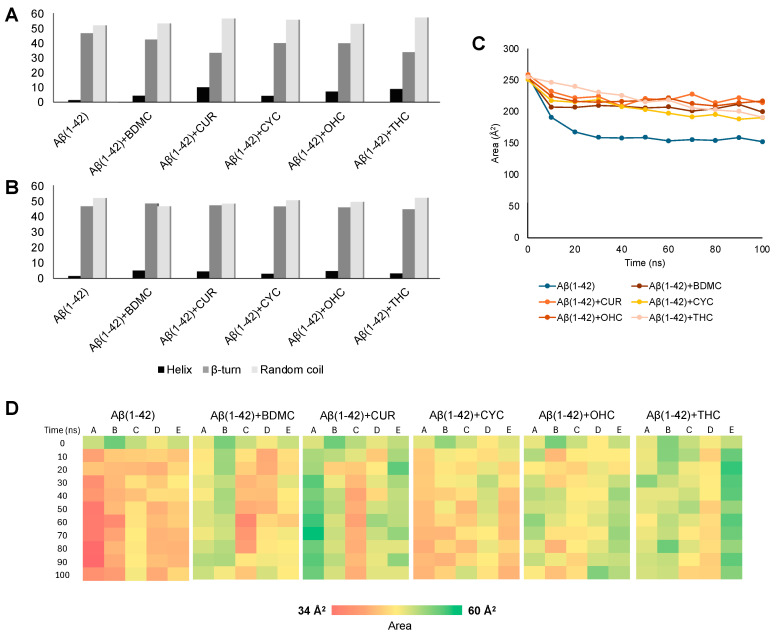
MD results of Aβ(1–42) in complex with curcuminoids (BDMC, CUR, CYC, OHC, THC). Secondary structure prediction of the five amyloid monomers in the presence of the curcuminoids (**A**) in the “unbiased simulation” and (**B**) in the “biased simulation”. (**C**) Plot of the solvent-accessible surface area of all peptides in the presence of the curcuminoids as a function of the simulation time from the “unbiased simulation”. (**D**) Heatmap of the solvent-accessible surface area for each of the five Aβ(1–42) monomers (chains A–E) in the presence of the curcuminoids as a function of the simulation time from the “unbiased simulation”. The colour code in the bar below represents the solvent exposure, ranging from the smallest area measured (34 Å^2^) to the largest (60 Å^2^).

## Data Availability

Data are contained within the article.

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
