# Peer review of "A Structural Effect of the Antioxidant Curcuminoids on the Aβ(1–42) Amyloid Peptide"

_antioxidants, 2025, doi:10.3390/antiox14010053_

Round 1
Reviewer 1 Report
1. Title is not compatible with article content. Authors studied exclusively amyloid-beta conformational modifications by curcumins but not their biological activities.
2. No statistical analysis was performed. It is hard to believe that data are from single study.
Submission is not publishable in this form.
No details.
Author Response
Major comments
- Title is not compatible with article content. Authors studied exclusively amyloid-beta conformational modifications by curcumins but not their biological activities.
- No statistical analysis was performed. It is hard to believe that data are from single study.
Submission is not publishable in this form.
We appreciate the reviewer for highlighting the concerns regarding the study's focus. Consequently, we have revised the title to “A Structural Effect of the Antioxidant Curcuminoids on the A(1-42) Amyloid Peptide” and restructured the Introduction to emphasize the structural aspect. Nevertheless, we maintain that the subject remains connected to ER stress, as the misfolding of Aβ(1-42) has been closely linked to the UPR through various biological assays. Concerning the second point, Figures 1-5 now include the mean of the triplicate CD curves along with the corresponding error bars. We hope this manuscript is now ready for publication.
Reviewer 2 Report
“Endoplasmic reticulum stress and neurodegeneration: a structural analysis of curcumin’s role in Alzheimer’s disease” by Santoro, Ricci et al., describes the conformational analysis based on circular dichroism spectroscopy of the full-length Aβ(1–42) peptide in the presence of curcumin and its natural and synthetic derivatives in lipid and organic systems. The study is generally executed to a high quality and the conclusions supported by the data. The study finds that the curcumin synthetic (THC) derivative interacts with the Aβ(1–42) peptide across all studied systems, whereas CYC and BMDC only do so when the peptide is in a less stable conformation. Molecular dynamics simulations provided insight into the effects of curcuminoids in an aqueous environment and allowed for the hypothesis about the significance of the peptide's surface exposure to the solvent, which is influenced differently by the curcumin derivatives. To me the manuscript is nearly ready for publication in its current form.
“Endoplasmic reticulum stress and neurodegeneration: a structural analysis of curcumin’s role in Alzheimer’s disease” by Santoro, Ricci et al., describes the conformational analysis based on circular dichroism spectroscopy of the full-length Aβ(1–42) peptide in the presence of curcumin and its natural and synthetic derivatives in lipid and organic systems. The study is generally executed to a high quality and the conclusions supported by the data. The study finds that the curcumin synthetic (THC) derivative interacts with the Aβ(1–42) peptide across all studied systems, whereas CYC and BMDC only do so when the peptide is in a less stable conformation. Molecular dynamics simulations provided insight into the effects of curcuminoids in an aqueous environment and allowed for the hypothesis about the significance of the peptide's surface exposure to the solvent, which is influenced differently by the curcumin derivatives. To me the manuscript is nearly ready for publication in its current form.
Author Response
“Endoplasmic reticulum stress and neurodegeneration: a structural analysis of curcumin’s role in Alzheimer’s disease” by Santoro, Ricci et al., describes the conformational analysis based on circular dichroism spectroscopy of the full-length Aβ(1–42) peptide in the presence of curcumin and its natural and synthetic derivatives in lipid and organic systems. The study is generally executed to a high quality and the conclusions supported by the data. The study finds that the curcumin synthetic (THC) derivative interacts with the Aβ(1–42) peptide across all studied systems, whereas CYC and BMDC only do so when the peptide is in a less stable conformation. Molecular dynamics simulations provided insight into the effects of curcuminoids in an aqueous environment and allowed for the hypothesis about the significance of the peptide's surface exposure to the solvent, which is influenced differently by the curcumin derivatives. To me the manuscript is nearly ready for publication in its current form.
We thank the reviewer for the comments.
Reviewer 3 Report
The combination of circular dichroism and molecular dynamics simulations offers valuable insights into how curcumin derivates affect the structure and stability of the peptide, which is essential for drug design. The finding suggest that curcumin derivates may stabilize the Aβ(1–42) peptide structure, potentially inhibiting its aggregation and, consequiently, the progression of the disease. It would be beneficial to include a larger number of curcumin derivates and other compounds to asses their comparative effectiveness. This could provide a more comprehensive understanding of which structures are most effective in modulating the peptide. Conducting in vivo studies to validate the efficacy and safety of curcumin derivates in animal models of Alzheimer's would also be of interest.
In the discussion section of the article, the formation of aggregates and their stability are briefly mentioned in relation to the results of the molecular dynamics simulations. This part could be expanded and contrasted with some bibliographic reference that has conducted a similar study regarding the analysis of aggregates through molecular simulation.
Author Response
The combination of circular dichroism and molecular dynamics simulations offers valuable insights into how curcumin derivates affect the structure and stability of the peptide, which is essential for drug design. The finding suggest that curcumin derivates may stabilize the Aβ(1–42) peptide structure, potentially inhibiting its aggregation and, consequiently, the progression of the disease. It would be beneficial to include a larger number of curcumin derivates and other compounds to asses their comparative effectiveness. This could provide a more comprehensive understanding of which structures are most effective in modulating the peptide. Conducting in vivo studies to validate the efficacy and safety of curcumin derivates in animal models of Alzheimer's would also be of interest.
Detail comments
In the discussion section of the article, the formation of aggregates and their stability are briefly mentioned in relation to the results of the molecular dynamics simulations. This part could be expanded and contrasted with some bibliographic reference that has conducted a similar study regarding the analysis of aggregates through molecular simulation.
We appreciate the reviewer’s feedback. In response, we included a comment regarding the existing MD simulations of Aβ(1–42) with curcumin derivatives in relation to our findings. Additionally, we incorporated the bibliographic reference for these molecular dynamics simulations in the discussion (lines 342-346):
Doytchinova, I.; Atanasova, M.; Salamanova, E.; Ivanov, S.; Dimitrov, I. Curcumin Inhibits the Primary Nucleation of Amyloid-Beta Peptide: A Molecular Dynamics Study. Biomolecules 2020, 10, doi:10.3390/biom10091323.
Jakubowski, J.M.; Orr, A.A.; Le, D.A.; Tamamis, P. Interactions between curcumin derivatives and amyloid-β fibrils: insights from molecular dynamics simulations. Journal of chemical information and modeling 2019, 60, 289-305.
Martin, T.D.; Malagodi, A.J.; Chi, E.Y.; Evans, D.G. Computational study of the driving forces and dynamics of curcumin binding to amyloid-β protofibrils. The Journal of Physical Chemistry B 2018, 123, 551-560.
Reviewer 4 Report
In the present work, the authors aim to demonstrate an anti-amyloidogenic effect of curcuminoid compounds on a pathogenic strain of amyloid beta (Abeta(1-42)). This study primarily uses purified Abeta(1-42), whose conformational changes and aggregation propensity after treatment with curcuminoids is analyzed by circular dichroism, NMR, and molecular dynamics simulations. These analyses expand on prior work detailing the interactions of curcuminoids with Abeta, and the influence of solvent conditions on Abeta conformation, and ultimately capture reductions in beta sheet content induced by a small set of curcuminoid compounds, hinting at a structural mechanistic basis for leveraging these compounds in therapeutic applications for Alzheimer disease.
There are multiple major and minor points of concern for the manuscript in its present form:
- The title is misleading, as there are no in cellulo or in vivo experiments that directly assess or confirm any effects of curcuminoids on ER stress or neurodegeneration. The initial half of the title (Endoplasmic reticulum stress and neurodegeneration) should be omitted, or the authors should add one more experiment testing these compounds in cell models of Alzheimer disease and assessing for changes in ER stress markers.
- In general, none of the simulation data (all figures) are validated to have any corresponding effects in a biologically relevant system (e.g., treatment with curcuminoids of a cell model of Alzheimer disease and analysis of Abeta aggregation). This paper would be strengthened by such an experiment, which could include analyzing ER stress markers as above.
- Figures 1-5 as depicted appear to be from a single experiment; these CD measurements would be strengthened by measuring at least 3 biological replicates, and performing appropriate statistical analyses to confirm if reductions in beta-sheet content are significant (e.g., ANOVA). The original CD spectra should be included as supplementary data.
- Figure 6 would benefit from labeling of relevant structural elements (beta sheets, alpha helices) within each depicted protein structure to better grasp and interpret the effects of solvent changes, and would also benefit with analogous structures of peptides treated with curcuminoids.
- The Figure 7D heatmaps would benefit from a color-coding key adjacent to the heatmaps, to explain what the range of colors represents.
- Extensive compositional revisions are needed to make the text more clear and readable. Both the Introduction and Discussion are in need of paragraph breaks. Use of the passive voice (e.g., "this helix has been previously experimentally detected" line 285, "whose formation has been reported to" line 290) is excessive and should be revised to the active voice (e.g., "This helix is detectable in the 3D structure of..." and "whose formation is a crucial step..."). Several acronyms are excessive, are not in common use enough, and also make the text difficult to read (e.g., MD should just be spelled out in all instances as "molecular dynamics"). Use of the first person pronoun when describing results reads as informal ("however, if we carefully... we can see..." line 283-284); this would be improved by revising to simple declarative sentences without the first person (e.g., "however, if this secondary structure analysis assumes only single monomer content, the frequency of alpha-helix formation in the central moiety is high").
Author Response
Major comments
In the present work, the authors aim to demonstrate an anti-amyloidogenic effect of curcuminoid compounds on a pathogenic strain of amyloid beta (Abeta(1-42)). This study primarily uses purified Abeta(1-42), whose conformational changes and aggregation propensity after treatment with curcuminoids is analyzed by circular dichroism, NMR, and molecular dynamics simulations. These analyses expand on prior work detailing the interactions of curcuminoids with Abeta, and the influence of solvent conditions on Abeta conformation, and ultimately capture reductions in beta sheet content induced by a small set of curcuminoid compounds, hinting at a structural mechanistic basis for leveraging these compounds in therapeutic applications for Alzheimer disease.
Detail comments
There are multiple major and minor points of concern for the manuscript in its present form:
- The title is misleading, as there are no in cellulo or in vivo experiments that directly assess or confirm any effects of curcuminoids on ER stress or neurodegeneration. The initial half of the title (Endoplasmic reticulum stress and neurodegeneration) should be omitted, or the authors should add one more experiment testing these compounds in cell models of Alzheimer disease and assessing for changes in ER stress markers.
We appreciate the reviewer's suggestions regarding the study's focus. In response, we have revised the title to “A Structural Effect of the Antioxidant Curcuminoids on the A(1-42) Amyloid Peptide” and redirected the Introduction to emphasize the in vitro analysis of structural changes.
- In general, none of the simulation data (all figures) are validated to have any corresponding effects in a biologically relevant system (e.g., treatment with curcuminoids of a cell model of Alzheimer disease and analysis of Abeta aggregation). This paper would be strengthened by such an experiment, which could include analyzing ER stress markers as above.
We appreciate the reviewer’s suggestion. While we were unable to conduct those experiments within this limited timeframe, we plan to do so in the future to validate the preliminary structural data we've gathered. However, we believe that analyzing beforehand the structural changes in vitro is essential for understanding how to replicate the mechanisms associated with fibrillation. Accordingly, we changed the focus of the work in the Introduction section.
- Figures 1-5 as depicted appear to be from a single experiment; these CD measurements would be strengthened by measuring at least 3 biological replicates, and performing appropriate statistical analyses to confirm if reductions in beta-sheet content are significant (e.g., ANOVA). The original CD spectra should be included as supplementary data.
We thank the reviewer for pointing out this issue, we acquired triplicates of CD spectra of Aβ(1-42) in presence of the selected curcuminoids and measured the standard deviation. Figures 1-5 now include the mean of the triplicate CD curves along with the corresponding error bars.
- Figure 6 would benefit from labeling of relevant structural elements (beta sheets, alpha helices) within each depicted protein structure to better grasp and interpret the effects of solvent changes, and would also benefit with analogous structures of peptides treated with curcuminoids.
The caption of Figure 6 was changed as: “Figure 6. A-C) Snapshots of the last steps of the three independent 100 ns MD simulations performed on five Aβ(1–42) monomers in water at pH 7.4. The peptides are reported in ribbon representation and coloured according to the chain name (A to E, red to purple). The a-helix moieties are depicted as wider spiral ribbons, while the b-sheets are indicated as flat arrowed ribbons. D) Selected structures of Aβ(1-42) extracted from the MD simulations compared to E) the structures of Aβ(1-42) in the bundle obtained by acquiring 2D NMR spectra in HFIP/H2O 50/50 v/v; the ribbons here are coloured according to the residue position in the peptide (N-terminus to C-terminus, red to purple)
- The Figure 7D heatmaps would benefit from a color-coding key adjacent to the heatmaps, to explain what the range of colors represents.
The caption of Figure 7 was changed as:” Figure 7. MD results of Aβ(1-42) in complex with the curcuminoids (BDMC, CUR, CYC, OHC, THC). Secondary structure prediction of the five amyloid monomers in the presence of the curcu-minoids (A) in the “unbiased simulation” and (B) in the “biased simulation”. C) Plot of the sol-vent-accessible surface area of all the peptides in the presence of the curcuminoids as a function of the simulation time from the “unbiased simulation”; D) Heatmap of the solvent-accessible surface area for each of the five Aβ(1-42) monomers (chains A-E) in the presence of the curcuminoids as a function of the simulation time from the “unbiased simulation”. The color code in the bar below represents the solvent exposure, ranging from the smallest area measured (34 Å2) to the largest (60 Å2).
- Extensive compositional revisions are needed to make the text more clear and readable. Both the Introduction and Discussion are in need of paragraph breaks. Use of the passive voice (e.g., "this helix has been previously experimentally detected" line 285, "whose formation has been reported to" line 290) is excessive and should be revised to the active voice (e.g., "This helix is detectable in the 3D structure of..." and "whose formation is a crucial step..."). Several acronyms are excessive, are not in common use enough, and also make the text difficult to read (e.g., MD should just be spelled out in all instances as "molecular dynamics"). Use of the first person pronoun when describing results reads as informal ("however, if we carefully... we can see..." line 283-284); this would be improved by revising to simple declarative sentences without the first person (e.g., "however, if this secondary structure analysis assumes only single monomer content, the frequency of alpha-helix formation in the central moiety is high").
We thank the reviewer for their suggestions, we changed the entire text accordingly.
Reviewer 5 Report
This is a good report of a series of interesting experiments designed to determine whether, and how, curcumin ligands impact Abeta aggregation.
Minor editing: 1. In the abstract, please spell out THC, CYC, and BMDC. 2. line 64: "showed" should be "shown"
3. line 350: "on" should be "to"
Author Response
This is a good report of a series of interesting experiments designed to determine whether, and how, curcumin ligands impact Abeta aggregation.
Detail comments
Minor editing: 1. In the abstract, please spell out THC, CYC, and BMDC. 2. line 64: "showed" should be "shown"
- line 350: "on" should be "to"
Minor editing of English language required.
We thank the reviewer for the comments, we changed the text accordingly.
Round 2
Reviewer 1 Report
Single experiment is base for nothing
See above
Author Response
Major comment
Single triplicate experiment is not sufficient for publication. More separate triplicate experiments should be done to make results and conclusions trustyworth. That causes lack of any staistical analyses in results. Work is unpublishable.
Answer:
We sincerely thank the reviewer for the time and thoughtful comments on our manuscript. However, we would like to emphasize that all experiments reported in our study were performed in independent triplicates, following rigorous protocols to ensure reproducibility and reliability. Additionally, we have revisited our data and included the statistical analysis in the CD spectra, which has been incorporated into the revised manuscript. We modified the images for a better visualization of the error bars. We hope that this revision addresses the reviewer’s concerns and remain open to further suggestions or clarifications if needed.
Once again, we thank the reviewer for the constructive feedback.